# Machine Learning-Based Front Detection in Central Europe

Bogdan Bochenek [1,*], Zbigniew Ustrnul [1,2], Agnieszka Wypych [2] and Danuta Kubacka [1]

1   Institute of Meteorology and Water Management—National Research Institute, 01-673 Warsaw, Poland; zbigniew.ustrnul@uj.edu.pl (Z.U.); danuta.kubacka@imgw.pl (D.K.)
2   Department of Climatology, Jagiellonian University, 31-007 Kraków, Poland; agnieszka.wypych@uj.edu.pl
*   Correspondence: bogdan.bochenek@imgw.pl

**Abstract:** Extreme weather phenomena such as wind gusts, heavy precipitation, hail, thunderstorms, tornadoes, and many others usually occur when there is a change in air mass and the passing of a weather front over a certain region. The climatology of weather fronts is difficult, since they are usually drawn onto maps manually by forecasters; therefore, the data concerning them are limited and the process itself is very subjective in nature. In this article, we propose an objective method for determining the position of weather fronts based on the random forest machine learning technique, digitized fronts from the DWD database, and ERA5 meteorological reanalysis. Several aspects leading to the improvement of scores are presented, such as adding new fields or dates to the training database or using the gradients of fields.

**Keywords:** weather fronts; machine learning; random forest; ERA5

## 1. Introduction

Extreme weather conditions, due to their influence on human safety and life, and also on the environment and economy, are now under detailed investigation worldwide. The Intergovernmental Panel on Climate Change's (IPCC) Sixth Assessment Report [1] clearly states that there has been an increase in both the frequency and intensity of extreme events in recent years, which only confirms that we should try to understand and predict them better so that we can react appropriately. Some of the most dangerous extreme events are those related to atmospheric dynamic processes—e.g., hail, thunderstorms, and mesoscale convective systems [2–4]. These usually occur when two air masses with different physical properties collide. Rapid changes in temperature and humidity cause short-term, but very intense, weather phenomena. These layers separating air masses of different origins—characterized by a narrow layer of high-temperature gradient, density, and wind direction changes—are called weather fronts, as introduced by Bjerknes and Solberg [5]. This concept allows us to explain various weather processes and events and has become a brief way to communicate about them that is still used in synoptic meteorology [6–8]. Frontal systems have for a long time been acknowledged as the main driving force for extreme precipitation in midlatitudes [9–11]. Furthermore, anomalies along a frontal feature can generate instability, such as frontal-wave growth and cyclogenesis; secondary cyclones or even cyclone families can form, with the potential to cause high-impact weather [11,12].

Recent studies on frontal systems climatology confirm an increase in the number of fronts, especially strong, active fronts that are more likely to be linked to extreme weather [13,14]. Therefore, further research on front detection, spatiotemporal variability, forecast, and future projections is essential. Nevertheless, almost a hundred years after establishing the term, meteorologists still have not established a clear definition of a front. The most common (AMS Glossary) definition describes a front as the interface or transition zone between two air masses of different densities [15]. As temperature is the main regulator of atmospheric density, it is also regarded as the fundamental determinant. Nevertheless, many other features may characterize a front. The main problems are the

horizontal and vertical scale of fronts, but there is also a list of different meteorological variables that have to be taken into account [12]. The lack of precision in the definition of a weather front means that there is still no satisfactory method to determine the position of fronts. Both manual, subjective methods and also more automatic, objective methods are used. The drawing of synoptic-scale fronts on weather maps is an everyday duty of forecasters in many national weather services. These maps, in a semi-digital form, are available online [16–18]. The methodology and procedures for the drawing of maps with fronts differ between centers and also between forecasters, as the process itself is very subjective. As a consequence, the same meteorological situation can be visualized in many different ways.

In terms of manual methods, forecasters' subjective decisions are the most important aspect. In some cases, fronts are drawn only when there is an evident difference in meteorological conditions in some part of the atmosphere, while other researchers tend to mark fronts or quasi-fronts when there is some instability or small difference in the state of the atmosphere. There is a significant difference in front numbers, and even front forms, when one compares archived weather maps with present ones, published on different websites and bulletins [19–21]. At present, there are many convergence lines and secondary fronts drawn, which could not have been found 20–30 years before. Therefore, there is a problem with the lack of homogenous data.

The current situation, as presented above, as well as opportunities with new methods, techniques, and extended databases have led scientists to seek new ways to determine and analyze weather fronts. Some recent publications are devoted to this subject with the use of different input data [22,23].

Objective approaches for determining the positions of weather fronts can be divided into methods using frontal location functions and those using machine learning techniques. A thermodynamic definition of a front was proposed by Hewson [24], where any gridded meteorological dataset can be used to draw fronts automatically, without forecaster intervention. This method has been used, as well as slightly modified, improved, and updated, in a number of studies [8,10,12,25], with the conclusion being that it helps to increase forecasters' productivity. Another method is based on wind shift and acceleration [26] and has also been used in other studies [27]. Comparing the two methods, Hope et al. [28] found them equivalent, while, for Schemm et al. [29], the thermodynamic method was found to be better for midlatitude weather systems, while the wind method was better for regions with strong convergence or wind shear [11].

The second, more recent approach is related to machine learning techniques. Several recent publications have addressed this problem with the use of deep learning or deep convolutional neural networks. Biard and Kunkel [30] proposed the DL-FRONT algorithm, which uses labeled front datasets from the National Weather Service (NWS) Coded Surface Bulletin [31] and a two-dimensional convolutional neural network to produce objective front localizations over North America, detecting nearly 90% of the manually analyzed fronts. Deep convolutional neural networks were used by Liu et al. [32] for the detection of extreme climate events, such as weather fronts, tropical cyclones, and atmospheric rivers.

In this work, we propose a novel method for detecting weather fronts over Central Europe based on the digitized locations of fronts from Deutscher Wetterdienst (DWD) maps using the ArcGIS software, ERA5 reanalysis, and the random forest machine learning technique.

## 2. Materials and Methods

### 2.1. Study Area

The study area covers Central Europe, which is meteorologically and synoptically well-recognized by authors. This is the region located in the center of the European continent in a temperate interim climate zone where different air masses often mix together, especially those from the west (different types of polar maritime air masses) with those from the east (polar continental air masses). The occurrence of weather fronts is directly related to these

airflows, with a mean frequency of about 40% of the days in a year [13,33], and often they are characterized as very active. Often, changes in weather conditions are related to the high temporal frequency of these weather fronts. On many occasions, this leads to the occurrence of extreme weather conditions, such as thunderstorms with tornadoes and bow echoes in the summer, or high wind speeds with heavy snowfall in the winter [2–4,10].

Therefore, Central Europe is a good region for study and is often considered for testing. Additionally, the study area is geographically complex, with lowlands, highlands, mountains, and marine areas, and reflects different topographic conditions for the formation and transformation of weather fronts and their role in weather conditions.

All analyses were performed for the geographical region 5°–30° W and 45°–60° N.

### 2.2. Digitalization of DWD Front Maps

Weather fronts from the DWD are available in the form of GIF maps. The procedure for the digitalization of the geographical positions of fronts was invented in the ArcGIS environment. An example of one day is presented in Figure 1. Although all types of fronts were separated with this procedure, in this study we chose to use all of them together (the position of any kind of front is marked by a 1 in our dataset, while a 0 indicates the lack of a front). The whole database consists of 24 months of digitized fronts (from 1 September 2017 to 31 August 2019, with one map for every day from 12 UTC) from DWD maps. Forty-two percent of all points were labeled as cold fronts, 34% as warm fronts, and 24% as occluded fronts.

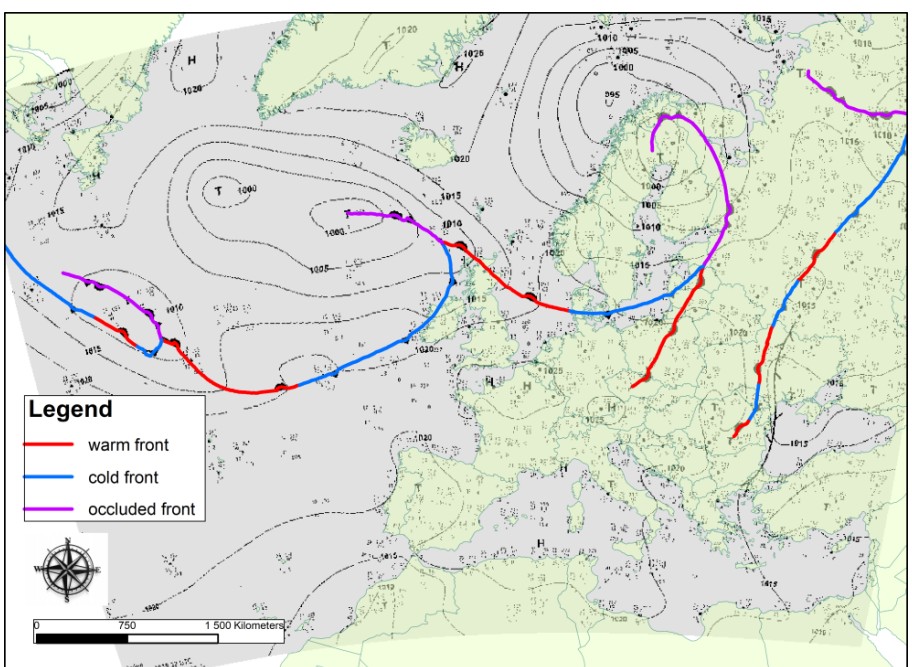

**Figure 1.** Example of the digitalization procedure.

### 2.3. Meteorological Reanalysis

The digitized weather fronts were used for further analysis based on data from an ERA5 meteorological reanalysis [34]. We used 4 pressure levels—925, 850, 750, and 500 hPa (Table 1)—and all 16 available meteorological fields that can indicate changes in weather conditions in the selected area. These levels were chosen after many attempts and discussions with forecasters from the IMGW-PIB Central Meteorological Office in Krakow over the last few years. Because of the significant variations in meteorological conditions near the ground, only 12 fields from the surface level were used in this analysis (Table 2). The correctness of this selection was confirmed in several situations, where fields and levels that were not relevant to the occurrence and transformation of weather fronts were excluded.

**Table 1.** ERA5 pressure level variables.

| Nr | Name | Abbreviation |
|----|------|--------------|
| 1 | divergence | d |
| 2 | fraction of cloud cover | cc |
| 3 | geopotential | z |
| 4 | mass mixing ratio | o3 |
| 5 | potential vorticity | pv |
| 6 | relative humidity | r |
| 7 | specific cloud ice water content | ciwc |
| 8 | specific cloud liquid water content | clwc |
| 9 | specific humidity | q |
| 10 | specific rain water content | crwc |
| 11 | specific snow water content | cswc |
| 12 | temperature | t |
| 13 | u-component of wind | u |
| 14 | v-component of wind | v |
| 15 | vertical velocity | w |
| 16 | vorticity | vo |

**Table 2.** ERA5 surface variables.

| Nr | Name | Abbreviation |
|----|------|--------------|
| 1 | 10 m u-component of wind | 10u |
| 2 | 10 m v-component of wind | 10v |
| 3 | 2 m temperature | 2t |
| 4 | skin temperature | skt |
| 5 | cloud base height | cbh |
| 6 | high cloud cover | hcc |
| 7 | low cloud cover | lcc |
| 8 | medium cloud cover | mcc |
| 9 | total cloud cover | tcc |
| 10 | mean sea level pressure | msl |
| 11 | total precipitation | tp |
| 12 | surface pressure | sp |

*2.4. Machine Learning*

The random forest method [35,36]—an ensemble machine learning method based on the construction of many decision trees that is widely used for many applications in meteorology [37–40], climatology [41,42], medicine [43,44], renewable energy [45–47], and many other fields—was used to build a model that combined meteorological parameters from the ERA5 dataset with the positions of fronts from digitized DWD maps. Since atmospheric conditions differ significantly between weather seasons in Central Europe, our analyses were performed separately for winter (DJF), spring (MAM), summer (JJA), and autumn (SON). In the first experiment, we trained the model from 1 to 30 January 2019, then examined different configurations for 31 January 2019. Finally, more general verification was performed for all days with fronts in the study area in January, April, July, and October. In addition, the impact of the length of the training period on the scores was examined. For example, 1 month of training data for days in January 2019 means all days from the same month; 3 months of training data for days in January 2019 means all the days from the same season (December 2018, January 2019, and February 2019); and 6 months of training data for days in January 2019 means all days from the same season and the same season of the previous year (December 2017, January 2018, February 2018, December 2018, January 2019, and February 2019).

### 2.5. Error Metrics

Standard metrics, such as probability of detection (POD [48]) and false alarm rate (FAR [49]) scores, were used to determine the impact of changing the length of the training period, adding surface fields to the data on pressure levels and the spatial sizes of fronts during the training process, and training with the values of the horizontal gradients of the meteorological fields.

## 3. Results

Several experiments were prepared to determine the best method for building a system to objectively determine the positions of weather fronts. The following subsections will show the results depending on the size of the fronts in testing and training; the differences in scores when pressure level fields were used with or without surface fields, using the horizontal gradients of meteorological fields in comparison to their original values; and the impact of the length of the training period.

### 3.1. Variable Importance

Since the random forest method enables us to look at the characteristics of the model that was built from the training dataset, a variable importance plot is presented in Figure 2. Out of the ten most important variables (we present only ten variables for the clarity of the plot), eight were from pressure level fields, and only two were from surface fields. The most important variable was the specific rain water content at 925 hPa and the second was total precipitation. There were also two other fields at 925 hPa (specific cloud liquid water content and specific humidity), and the specific cloud liquid water content was ranked in the top ten most important variables at three different pressure levels (925, 850, and 700 hPa). Another interesting result is related to the importance of geopotential at 700 and 500 hPa and the date of the model.

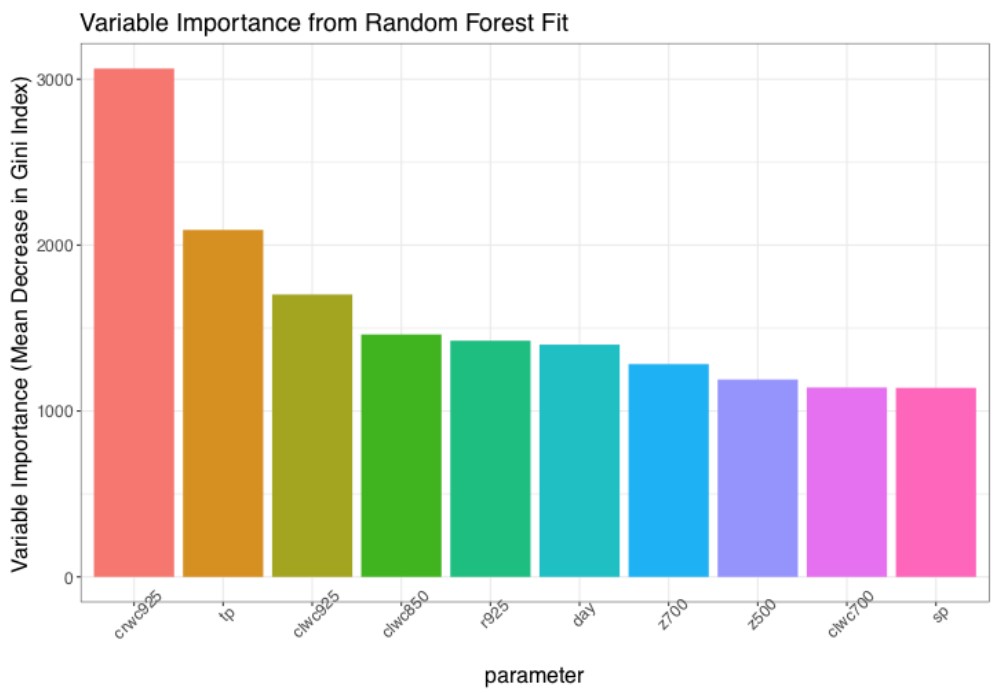

**Figure 2.** Variable importance plot.

### 3.2. Size of Fronts in Training and Testing

Weather maps show fronts as lines, where in reality they are bigger areas. Because there is no univocal definition of a front, different criteria were taken into consideration—e.g., a minimum extension of 500 km [29] and at least three contiguous grid points [8], or two or more neighboring grid points masked, in order to be considered a front [23]. That is

why we studied the optimal size of a front in our system. For every front point from the digitized database and the ERA5 data (both for surface and pressure level fields), we took into consideration neighboring grid points to test their optimal number.

Figure 3 shows an example of a situation from 31 January 2019, which is presented in its original form from the DWD database in Figure 4. The green areas show hit events by the system, the red areas indicate miss events, while false alarms are presented as blue dots on the maps. When only one point is taken into consideration (Figure 3a), the system produces only miss events; in reality, on the DWD map, the front was located slightly to the north of the system's prediction. Adding more points to the analysis (Figure 3b–f) resulted in a better POD score. The most optimal configuration of this analysis is with four additional points for every front coordinate, with the POD being higher than the FAR (Figure 5).

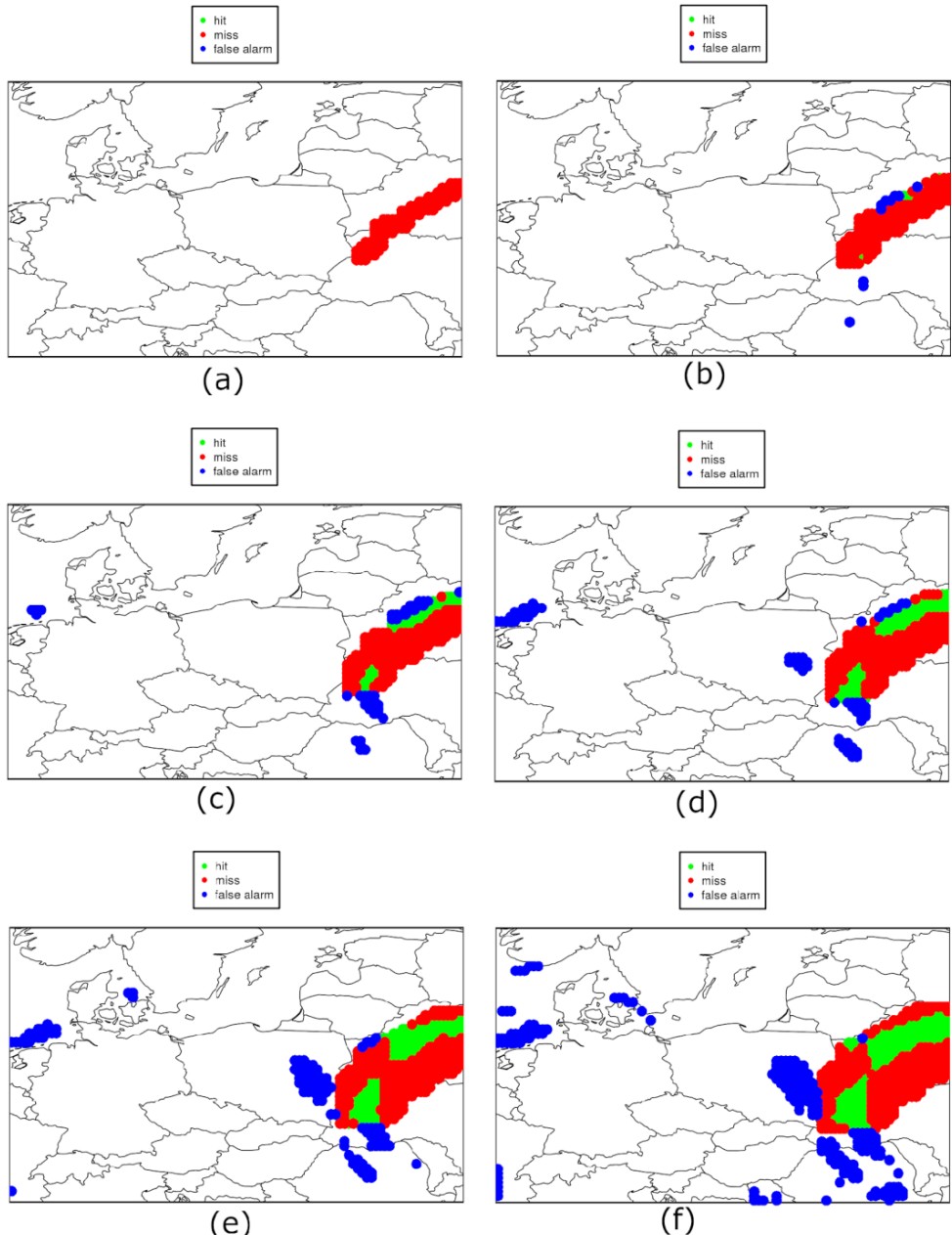

**Figure 3.** Results of the trained system with respect to the size of the front. Only one point of the front (**a**); additional one point (**b**); additional two points (**c**); additional three points (**d**); additional four points (**e**); additional five points (**f**).

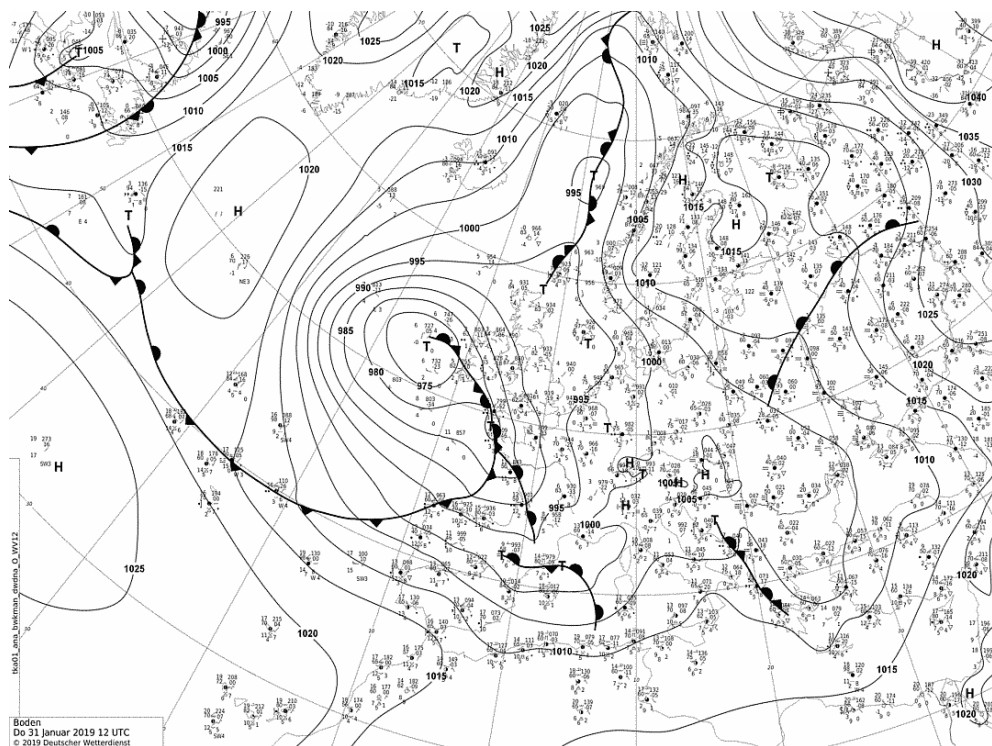

**Figure 4.** Original DWD map for 12 UTC, 31 January 2019.

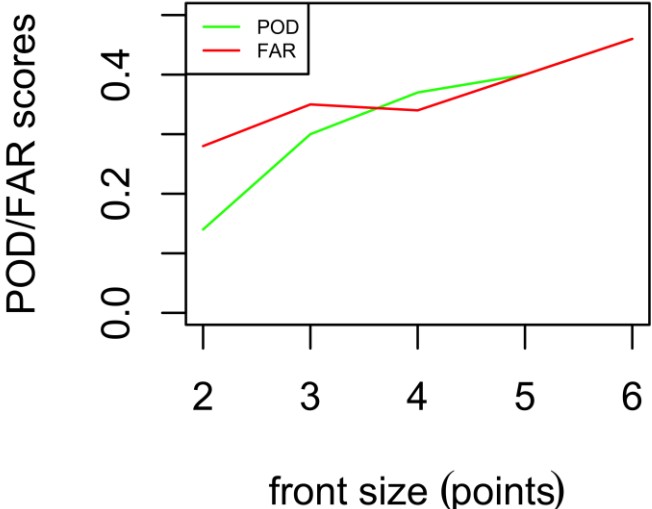

**Figure 5.** POD and FAR scores as a function of the front size.

### 3.3. Surface and Pressure Levels Fields

In the previous subsection, the training of the model was performed with both surface and pressure levels fields from an ERA5 reanalysis. Figure 6 shows the same configuration as presented in Figure 3d, but with results excluding surface fields. Removing the surface fields from the model decreased the POD by 40% and increased the FAR by 35% in this case. In further experiments, both surface and pressure level fields were considered.

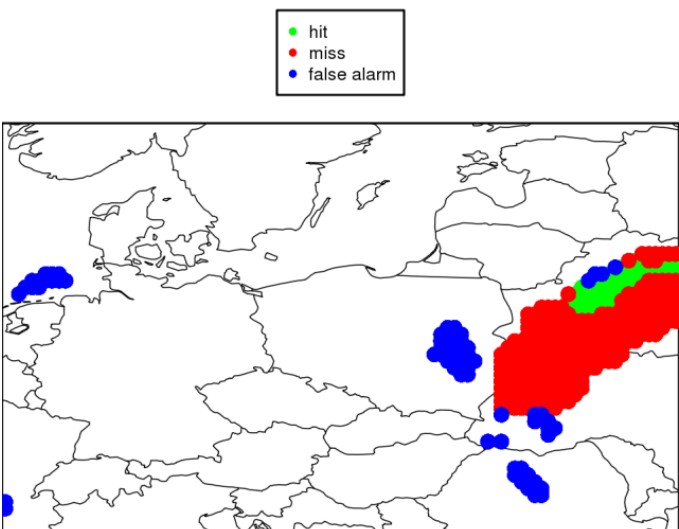

**Figure 6.** Results of the trained system with respect to removing the surface fields from the training database.

### 3.4. Gradients of Meteorological Fields

A front, separating air masses of different features, indicates a discontinuity of particular weather elements, especially air temperature, air pressure moisture, and wind (AMS). Therefore, their horizontal gradient is often used in front detection as one of the explanatory variables [23,50]. We included it in our approach as one of the model's adaptation procedures. The use of the horizontal gradients of the meteorological parameters instead of their values was tested, as well as the area taken into account for gradient calculation, which is presented in Figure 7. Figure 7a shows the previous results for the values of the meteorological fields, while Figure 7b–e present a gradient approach with the number of points used for calculation ranging from one to four. As is shown in Figure 8, the best results were obtained from gradients calculated with the use of one surrounding points. While the POD remained similar through the experiments, the FAR for this configuration was decreased.

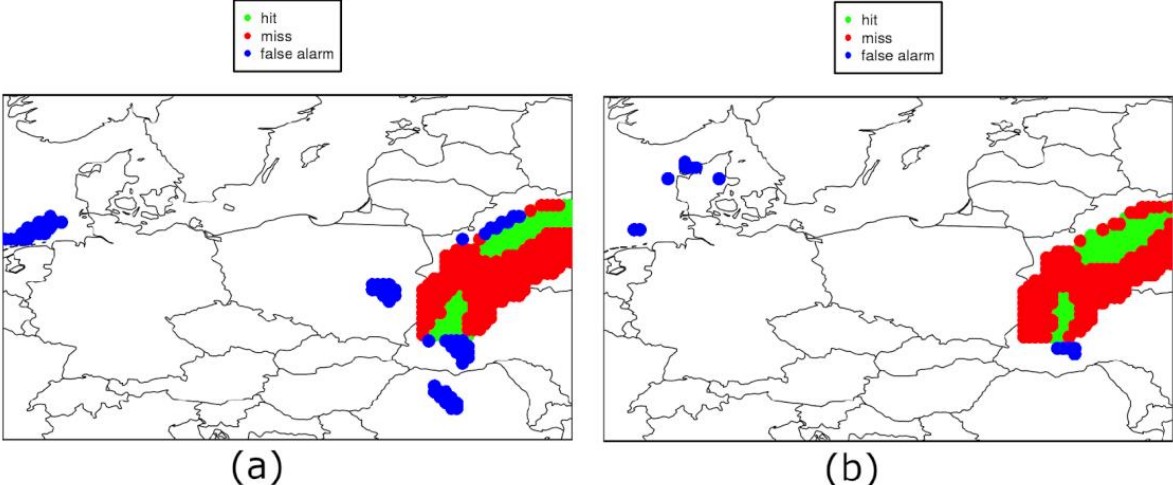

**Figure 7.** *Cont.*

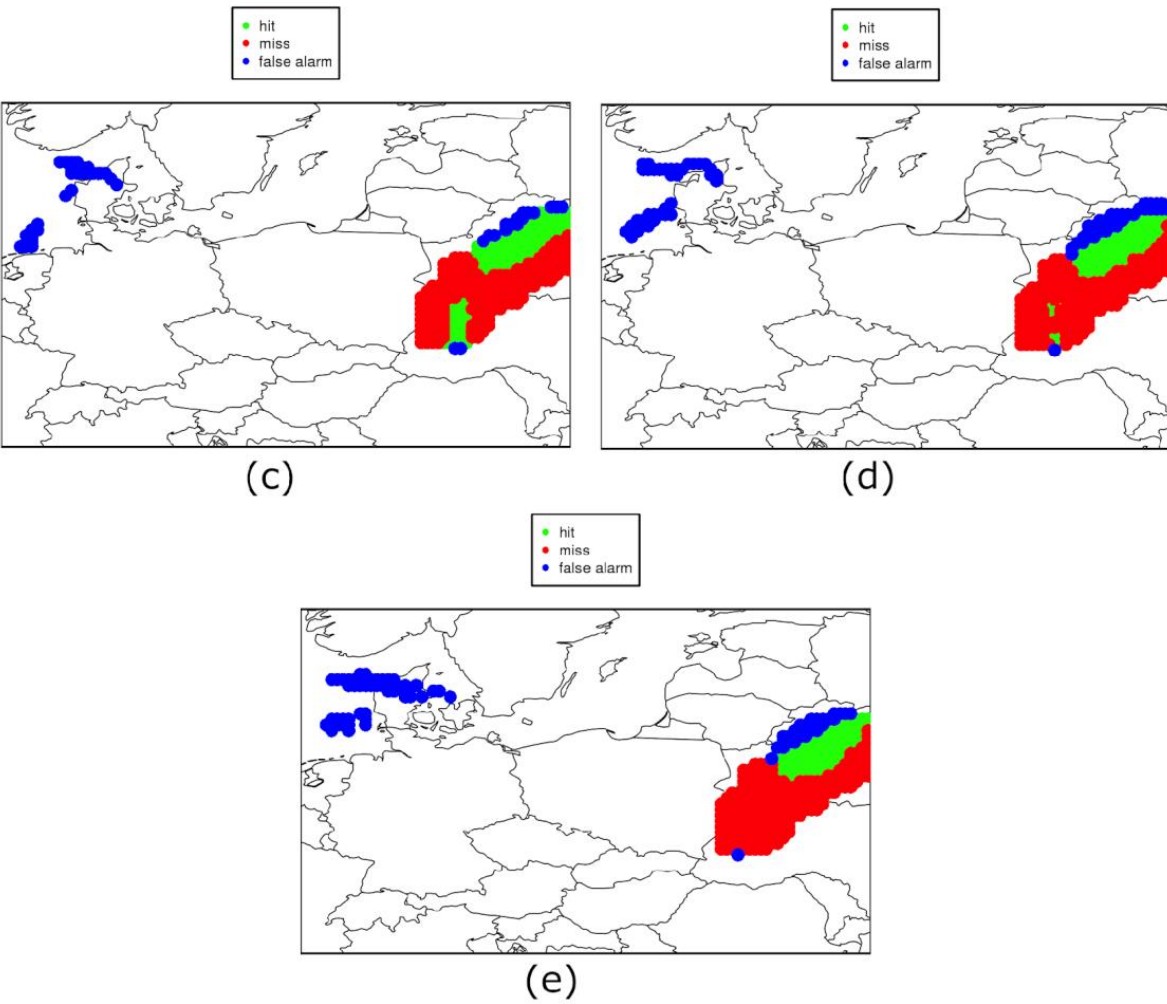

**Figure 7.** Results of the trained system with respect to using the values of meteorological parameters (**a**); the gradient of the meteorological fields calculated from one surrounding point (**b**); two surrounding points (**c**); three surrounding points (**d**); and four surrounding points (**e**).

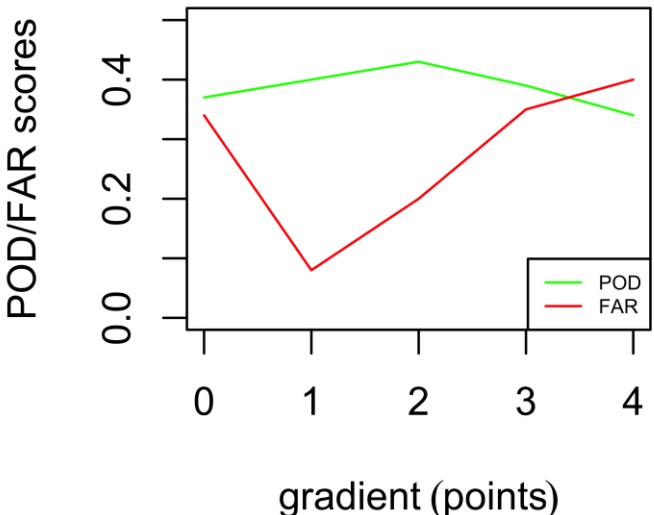

**Figure 8.** POD and FAR scores as a function of the number of points used for the calculation of the gradients of the meteorological fields.

### 3.5. Length of Training Period

The last examined improvement in our system was related to the length of the training period. As shown in many publications—i.e., by Floares et al. [51]—various scores can be significantly improved when the sample size of the dataset used for training the model is increased. A significant improvement of the FAR score can be achieved with 6 months of training (Figure 9c) compared to 1 month (Figure 9a). The biggest improvement is the reduction in points with false alarms, while the level of hits and misses remains similar.

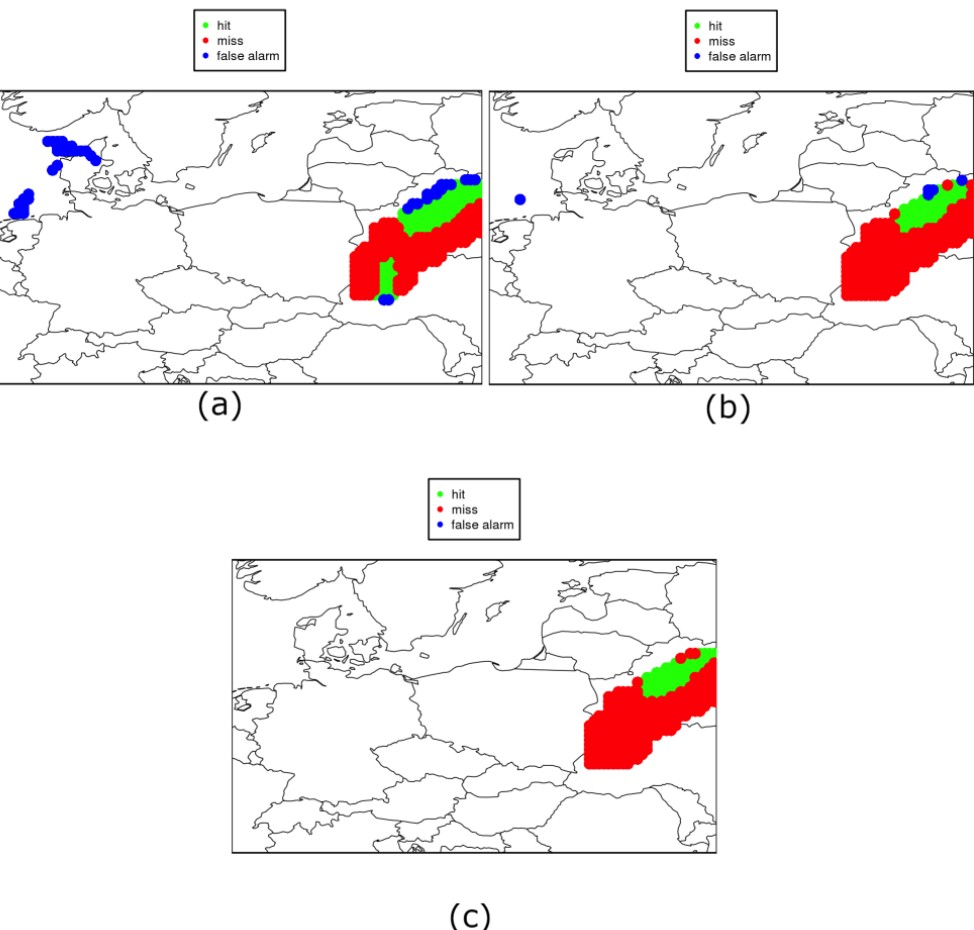

**Figure 9.** Results of the trained system with respect to the length of the training period. One month of training (**a**); three months of training (**b**); and 6 months of training (**c**).

All previous analyses have led to the conclusion that the best configuration for our model, which is based on ERA5 reanalysis and the random forest method, can be achieved by expanding the front size in the training dataset; using surface and pressure level fields with their gradients; and, possibly, using a longer period of training for the dataset.

### 3.6. Another Case Study

In the previous subsections, the results were presented for selected days in winter. Here, we show in Figure 10 a similar study for another day in winter and one day in summer 2019. For both days, there was a significant number of points with fronts in the region. In Figure 10a, a situation from 16 January 2019 is presented where there was an active warm weather front over Central Europe, related to a low-level pressure system with a center over southern Scandinavia, 24 h precipitation of up to 7 mm in northern Poland, and a few centimeters of fresh snow in the Tatra Mountains. The positions of the fronts, in this case, were correctly predicted by the model, especially in the north and central areas of the region. False alarms were mostly present over the coast of Germany, where there

was a warm sector between warm and cold fronts. Several missing values were recorded in the southeast of the region, where there was a weaker cold front drawn on the DWD weather map. Over the whole region, this situation was predicted rather correctly, with a POD equal to 55% and a FAR of 27%.

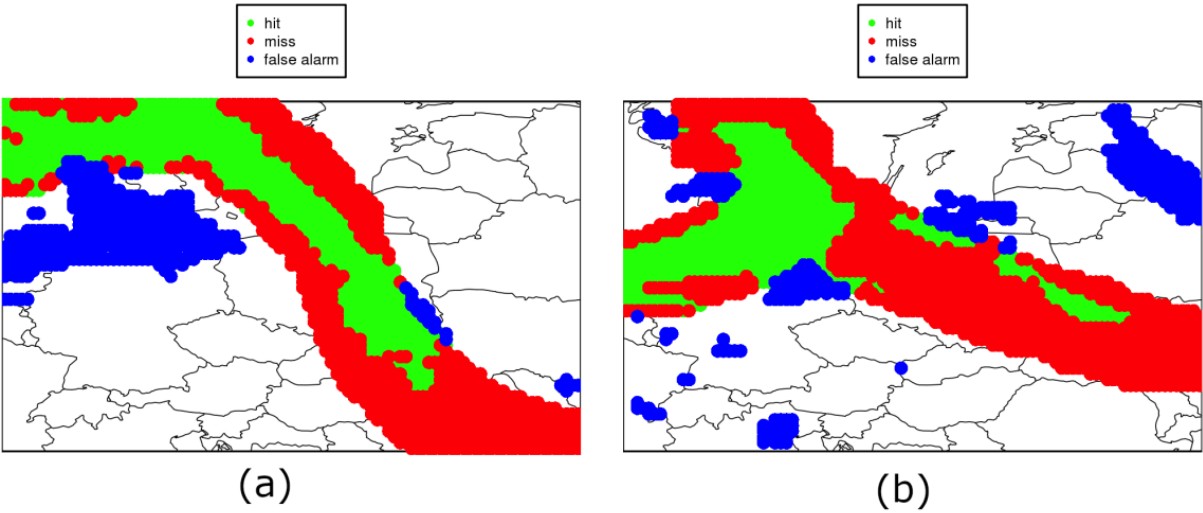

**Figure 10.** Results of the detection of fronts for 16 January 2019 (**a**) and 6 July 2019 (**b**).

The next situation in this subsection is from 6 July 2019 (Figure 10b), where there was, again, a low low-level pressure system with a center over southern Scandinavia, with an active cold front over the coast of Germany and a weaker warm front over Poland. The cold front was predicted very accurately, while the warm front predictions showed many missing values. Over the whole region, this situation was predicted rather correctly, with a POD equal to 50% and a FAR of 27%.

To better understand the model, the characteristics of the POD and FAR scores for every day in January 2019 are presented in Table 3. On several days, such as 1 January 2019 or 15 January 2019, the proposed method predicts front positions with a high POD and a low FAR, but on the other hand, several days show the opposite, such as 4 January 2019 or 6 January 2019. Figure 11 shows the meteorological conditions on IMGW-PIB weather maps for those days. During the days with a low accuracy from the model (Figure 11, top row), weather conditions were rather stable, with low-level systems present on the borders of the study area. For days with a high accuracy (Figure 11, bottom row), the meteorological situation was more dynamic, with more than one front passing through the center of the selected region. Similar tests were performed for other seasons, with the best results obtained for winter and autumn and an approximately 20% degradation of the POD and FAR in summer and spring—for clarity, these are not presented in this paper.

**Table 3.** POD and FAR score for days with fronts in January 2019.

| Date | POD | FAR |
|---|---|---|
| 1 January 2019 | 0.8 | 0.15 |
| 2 January 2019 | 0.19 | 0.17 |
| 4 January 2019 | 0.33 | 0.5 |
| 5 January 2019 | 0.37 | 0.2 |
| 6 January 2019 | 0.15 | 0.52 |
| 7 January 2019 | 0.22 | 0.2 |
| 8 January 2019 | 0.57 | 0.57 |
| 9 January 2019 | 0.09 | 0.25 |
| 10 January 2019 | 0.22 | 0.05 |

**Table 3.** *Cont.*

| Date | POD | FAR |
| --- | --- | --- |
| 11 January 2019 | 0.37 | 0.02 |
| 12 January 2019 | 0.52 | 0.31 |
| 13 January 2019 | 0.76 | 0.46 |
| 14 January 2019 | 0.25 | 0.21 |
| 15 January 2019 | 0.75 | 0.44 |
| 16 January 2019 | 0.56 | 0.26 |
| 17 January 2019 | 0.39 | 0.37 |
| 18 January 2019 | 0.08 | 0.27 |
| 23 January 2019 | 0.16 | 0.07 |
| 26 January 2019 | 0.61 | 0.25 |
| 27 January 2019 | 0.55 | 0.12 |
| 28 January 2019 | 0.16 | 0.29 |
| 30 January 2019 | 0.19 | 0.04 |

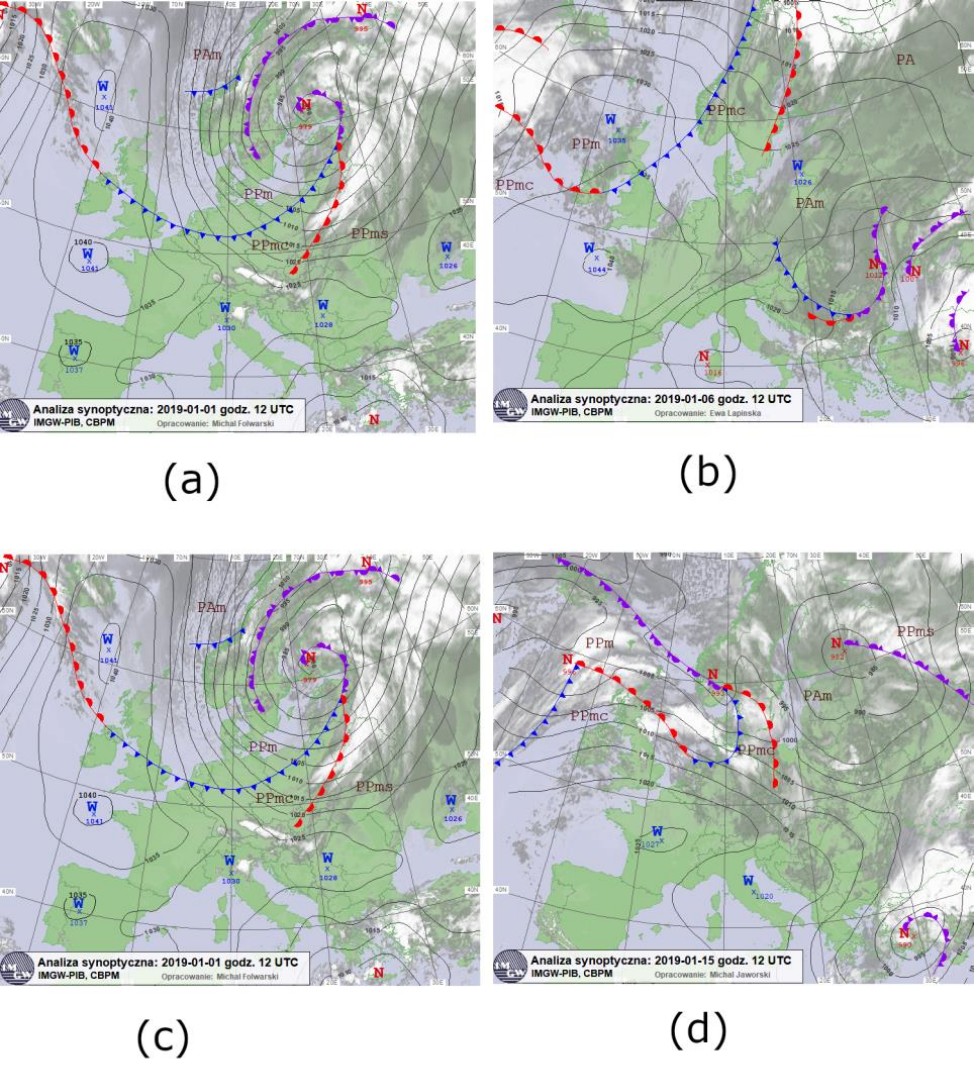

**Figure 11.** Meteorological conditions over Europe on IMGW-PIB weather maps from 4 January 2019 (**a**); 6 January 2019 (**b**); 1 January 2019 (**c**); and 15 January 2019 (**d**).

## 4. Discussion and Conclusions

In this study, we presented a new method for the objective determination of weather front positions with the use of a digitization procedure from weather maps and the random forest method. We have shown that, with a sample of digitized maps, we can train a machine learning model into a useful tool for the climatological analysis of fronts and for everyday forecasting duties. Using a substantive approach, we have confirmed the advantage of treating fronts as broader regions rather than as frontal lines, as well as using the horizontal gradients of meteorological fields rather than their raw values. Similar to other applications of machine learning techniques, we have shown that with more data and a longer training period, models will achieve better results.

Our work, which is the result of several previous attempts, used novel meteorological databases such as ERA5 and modern statistical techniques such as machine learning methods to address problems with the objective determination of fronts using the example of Central Europe. Some of the case studies that we presented showed that the proposed model works better with active fronts, with a POD of around 0.8 and a FAR of 0.15, but that it has problems with situations where fronts are vanishing and meteorological situations are rather stable, with a POD and FAR of around 0.2. We believe that, even if they are already promising, these results can be further improved.

The presented method will be developed using more data for training. The authors also hope that it will be possible to distinguish the different types of fronts, which is not yet possible because of the small sample size of digitized fronts. This issue seems to be the most problematic. In the future, thanks to this method, it will be possible to develop front climatology for Central Europe. This method will also be useful for other applications, such as the determination and prediction of air mass positions.

**Author Contributions:** Conceptualization, B.B. and Z.U.; methodology, B.B. and Z.U.; software, B.B. and D.K.; validation, B.B., Z.U. and A.W.; formal analysis, B.B.; investigation, A.W.; resources, Z.U.; data curation, A.W. and D.K.; writing—original draft preparation, B.B.; writing—review and editing, Z.U. and A.W.; visualization, B.B. and D.K.; supervision, Z.U. and A.W.; project administration, Z.U.; funding acquisition, Z.U. All authors have read and agreed to the published version of the manuscript.

**Funding:** The study was performed as part of the research task "Development of forecasting methods to improve existing products and develop new application solutions, task 4. Implementation and development of methods of analysis and meteorological forecasting for the needs of renewable energy sources (S-6/2021)" and "Modern climate warming and its impact on environment, task 6.", financed by the Ministry of Science and Higher Education (Poland), statutory activity of the Institute of Meteorology and Water Management-National Research Institute in 2021.

**Institutional Review Board Statement:** Not applicable.

**Informed Consent Statement:** Not applicable.

**Data Availability Statement:** Publicly available datasets were analyzed in this study. These data can be found here: https://danepubliczne.imgw.pl/datastore (accessed on 1 August 2021), http://www1.wetter3.de/ (accessed on 1 August 2021).

**Acknowledgments:** Authors would like to thank the operational weather forecasters from IMGW-PIB Central Meteorological Office in Krakow for many fruitful discussions.

**Conflicts of Interest:** The authors declare no conflict of interest.

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
