# Peer review of "Machine Learning-Based Front Detection in Central Europe"

_atmosphere, doi:10.3390/atmos12101312_

Round 1

Author Response

Reviewer 1

After almost thirty years in operational weather forecasting and twenty years of

academic research in the fields of synoptic - dynamic meteorology - synoptic and

dynamic climatology - physical mechanisms of extratropical cyclogenesis and

frontogenesis - Mediterranean tropical like cyclones, it was really challenging to read

a different philosophy on the identification of frontal activities.

Nevertheless, I don’t think this document meets the requirements of the journal in this

submitted form and major revision is required. More specifically:

  1. Title: ‘’AI-based fronts detection in Central Europe’’. What is ‘’AI’’?
  2.  

We decided to title our paper using the term AI (Artificial Intelligence), since recently several publications have already adopted it (e.g. AI based sensors for Forest Fires detection and AI based systems for Stopping the Spread, AI-based Pandemic Trend Analysis, Deep Learning and AI Based Systems to Detect Climate Calamities Before Occurrence and Raising Alarm When Needed). Nevertheless  we agree that a slight modification may contribute to its better understanding. The suggested title: “Machine learning-based front detection in Central Europe” has been applied. Thank you for that suggestion.

  1. b) I suppose the authors refer to the detection of surface fronts and not

upper level frontal activities. Am I correct? If so, how do they explain the

variable importance plot and the ranking of the ten more significant

atmospheric variables which includes, in majority, variables from

pressure levels? I’m really confused.

The aim of this work was detection of surface fronts that are presented on DWD maps. Upper level fronts are not presented in synoptic bulletins. Surface fronts are in practice drawn based on surface observations and measurement from synoptic stations and teledetection data. Fronts have a direct impact on meteorological conditions and they have been analysed since they were introduced. But front detection is of course difficult. Surface front has its vertical structure and that is why we decided to use ERA5 data not only from near-surface level, but also from selected pressure levels. We can say that we used a broad spectrum of data to train our model to detect surface fronts, and we let the model “to decide” which parameters are the most important.

What is also important, using only surface data will be problematic in complex terrain, like we see on many occasions in the Carpathian Mountains, where upper levels are necessary to consider. That is why we also used fields on upper levels. Overall it was 80 fields both from near-surface and 4 pressure levels. 

  1. c) If I’m not wrong the method is based on the analysis of the surface fronts

from DWD. DWD surface frontal analyses are subjective, not like the

analyses from Met Office which are objective. Can we assure that the

analyses of DWD are always correct? If not, then the machine is trained

with wrong inputs on a number of cases.

First of all we are grateful for the comment. As far as we know, weather observers (forecasters) in DWD get suggestions by an algorithm and then modify/draw the map according to their meteorological experience. We are in contact with them to fully clarify the issue. However, we decided to use these maps because they are freely available for a long period of time and are prepared for Central Europe, which is the region of our study. Although it will be interesting to compare these results with similar experiments done with different sources of front’s locations. 

DWD maps are freely available for a long period of time and are prepared for Central Europe (with quite convenient map projection), which is the region of our study. Of course it will be interesting to compare these results with similar experiments done with different sources of front’s locations. Independently, in our daily practice we look at both types of maps (i.e. originated from DWD as well as the Met Office and most often they are compatible for our region. Some differences concern their range (length), which is the result of quite subjective assessment, which is related to the lack of objective criteria for distinguishing fronts.

And finally, the matter of the learning material is very important, if it works for one type of map, we see no obstacles to use other sources (e.g. Met Office or KNMI or those provided by Polish or Czech met services). In the manuscript, we want to convey the research method. The very source of the learning material is (in this case) slightly less important. Of course ‘the better is the better’, but it cannot be easily assessed whether the original materials are produced in an objective (automatic) or expert (subjective) manner.

  1. d) How can we take a clear picture of the whole procedure if each type of the

frontal zones is not examined separately? How can we put on the same

bag the special and individual characteristics of the cold, warm, occluded

fronts, stationary fronts? How can we train the machine to exclude

baroclinic zones and retain only real frontal activities, if we examine all

frontal objects together?

This is our first approach to this problem, so we decided to try to build models for all types of fronts together. This way we had more data to train our system, and we learned the weaknesses of it. We focused now more on the aspect of discontinuity of air masses in the atmosphere, but we have in our minds intention to improve our model. We even tried to train it on different types of fronts separately, but there was now improvement of scores so far, and we believe that this is because of the small sample size of the training database.

  1. e) Actually the authors specify the deficiencies in the last two sentences of

Section 4. I would like to read this work again, after the inclusion of those

two issues that the authors refer at the end of Section 4.

We are aware that building models for all types of front together have some limitations but as we stated in previous answers, our current database of available, digitized fronts seems to be too small to perform such an experiment now. We have prepared some first results but they are not satisfactional so far, and we are not able to prepare more fronts in a short time, since digitalization of them requires some time and effort. We plan to continue our research in this field in that direction in the future.

Reviewer 2 Report

Frontal systems, especially cold fronts, are very often associated with extreme weather in midlatitudes, and have been the focus of weather studies for many years. The climatology of frontal systems are important for understanding climate change and also for prediction of extreme weather. This manuscript investigates the detection of fronts using AI technique. The machine learning technique used in this study is a well-accepted novel technique. The manuscript can be accepted for publication in Atmosphere after minor revision. The writing of the manuscript is generally clear. But some places need clarification and changes. In the text and figure captions, some places have mixed-up writings and it is very difficult to understand the content. Below are some detailed comments:

Major issues:

  1. This study propose a novel method to detect fronts. In Section 3.1, it is stated that 24 months of data are used. However, in Section 2.4, it is not stated very clearly how much data is used for training and how much is used for testing. I think Section 3.1 should be combined with section 2.4, and the manuscript should clearly state how much data is used for training and how much is used for testing.

As can be seen in Section 3.6, the length of data used for training can be 1 month or 6 months. These information should also be presented in Section 2.4.

One thing I don’t understand is how much data you used for training for each season. In Section 2.4, it is mentioned that January, April, July, and October are studied. However, how could you have enough training data for each season when the length of training period is 6 months? Each season only have 6 months of data in two years!

  1. In Section 2.3, lines 4-7, “those levels are chosen after many tries and discussions with forecasters…”. Please provide some objective criteria why these levels are chosen. Or please discuss the reasons why these levels are chosen. Obviously these are typical levels used in weather analysis. So it is very reasonable to choose these levels in this study.

  1. It seems that the figure orders are all shifted in the text. Please make sure all the figure numbers are consistent in the text. For example, in the text of Section 3.3, “Fig. 2” should be “Fig. 3”?
  2. In Section 3.6, “Fig. 8c” and “Fig. 8a” should be “Fig. 9c” and “Fig. 9a”? Based on the results in Fig. 9, I can see that the length of the data set does not seem to affect the results very much. So please clarify this in Section 3.6.
  3. I think that the use of gradients for improvement of scores is very interesting. I wonder if previous studies used this kind of method. Please discuss if gradient is used in previous studies.

Minor issues:

  1. Please check grammar. For example, the title should be “AI-based front detection”?

Probably phrases like “fronts detection, fronts number, fronts forms, fronts size” should all be changed to “front **”.

  1. Line 6 in the first paragraph, “dynamic of atmospheric processes”, should be “atmospheric dynamic processes”?
  2. Second paragraph, first sentence, there are two “which” in one sentence. This sentence should be made shorter.
  3. the title of 3.7: another case studies, should be “study”
  4. Abbreviations such as IPCC, NWS, DWD, etc.. should have details.
  5. Figure captions of Fig. 3 and Fig. 9 seem to contain some errors.

Author Response

Frontal systems, especially cold fronts, are very often associated with extreme weather in midlatitudes, and have been the focus of weather studies for many years. The climatology of frontal systems are important for understanding climate change and also for prediction of extreme weather. This manuscript investigates the detection of fronts using AI technique. The machine learning technique used in this study is a well-accepted novel technique. The manuscript can be accepted for publication in Atmosphere after minor revision. The writing of the manuscript is generally clear. But some places need clarification and changes. In the text and figure captions, some places have mixed-up writings and it is very difficult to understand the content. Below are some detailed comments:

Major issues:

  1. This study propose a novel method to detect fronts. In Section 3.1, it is stated that 24 months of data are used. However, in Section 2.4, it is not stated very clearly how much data is used for training and how much is used for testing. I think Section 3.1 should be combined with section 2.4, and the manuscript should clearly state how much data is used for training and how much is used for testing.

As can be seen in Section 3.6, the length of data used for training can be 1 month or 6 months. These information should also be presented in Section 2.4.

One thing I don’t understand is how much data you used for training for each season. In Section 2.4, it is mentioned that January, April, July, and October are studied. However, how could you have enough training data for each season when the length of training period is 6 months? Each season only have 6 months of data in two years!

Thank you for that comment. The paragraph has been re-edited to  be more clear. We merged section 3.1 with 2.4 and added supplementary explanation of differences between 1, 3 and 6 months of training:

Finally, more general verification was done for all days with fronts in the study area in January, April, July and October. Also the impact of the length of training period on scores was examined. E.g. 1 month of training data for days in January 2019 means all the other days from the same month, 3 months of training data for days in January 2019 means all the other days from the same season (December 2018, January 2019 and February 2019) and 6 months of training data for days in January 2019 means all the other days from the same season and the same season from previous year (December 2017, January 2018, February 2018, December 2018, January 2019 and February 2019)

  1. In Section 2.3, lines 4-7, “those levels are chosen after many tries and discussions with forecasters…”. Please provide some objective criteria why these levels are chosen. Or please discuss the reasons why these levels are chosen. Obviously these are typical levels used in weather analysis. So it is very reasonable to choose these levels in this study.

As indicated in the manuscript, main synoptic vertical levels have been taken into account. They are also available in available maps, previously also published in synoptic bulletins.

The number of 80 meteorological variables from 5 levels were taken in the analyzes. One could take many more of them (from many other available levels) but in our understanding they would not help much. Anyway, these additional levels were created on the basis of models, the initial ones, especially several years ago, included only the main ones.

  1. It seems that the figure orders are all shifted in the text. Please make sure all the figure numbers are consistent in the text. For example, in the text of Section 3.3, “Fig. 2” should be “Fig. 3”?
  2. In Section 3.6, “Fig. 8c” and “Fig. 8a” should be “Fig. 9c” and “Fig. 9a”? Based on the results in Fig. 9, I can see that the length of the data set does not seem to affect the results very much. So please clarify this in Section 3.6.

Answer to point 3 and 4: Thank you for pointing this problem out. Figures now are in the correct order. We added also a clarification in Section 3.5 (previously it was Section 3.5)

  1. I think that the use of gradients for improvement of scores is very interesting. I wonder if previous studies used this kind of method. Please discuss if gradient is used in previous studies.

Using horizontal gradients of meteorological parameters was applied also in other studies describing objective methods of distinguishing front locations, e.g.  Nakamura et al. 1986 or Partiff et al. 2017. In this study we decided to include horizontal gradients as one of the model adaptation procedures. A short explanation has been included in the revised text. 

Minor issues:

  1. Please check grammar. For example, the title should be “AI-based front detection”?

Probably phrases like “fronts detection, fronts number, fronts forms, fronts size” should all be changed to “front **”.

    The revised version of the manuscript has been corrected by a native speaker. 

  1. Line 6 in the first paragraph, “dynamic of atmospheric processes”, should be “atmospheric dynamic processes”?

The revised version of the manuscript has been corrected by a native speaker. 

  1. Second paragraph, first sentence, there are two “which” in one sentence. This sentence should be made shorter.

The revised version of the manuscript has been corrected by a native speaker. 

  1. the title of 3.7: another case studies, should be “study”

The title has been corrected. 

  1. Abbreviations such as IPCC, NWS, DWD, etc.. should have details.

Abbreviations have been explained at their first appearance.

  1. Figure captions of Fig. 3 and Fig. 9 seem to contain some errors.

    They have been corrected. 

Reviewer 3 Report

Overall, good topic is chosen to be studied. Authors tried to do their best but significant issues exist and need to be improved before going to printing/accepting.

Decision: major revisions

major points:

  1. front needs to be defined and why this paper doesn't follow a front definition by AMS/WMO
  2. why the parameters selected for front discrimination and AI analysis
  3. why they are important, method is not clear.
  4. results are given for extended frontal regions but for same no precip, cloud water content, and wind are provided, this is very weak point for this work.
  5. some figs are note indicated correctly, check.
  6. conclusions are very weak and not given in an itemized way.

Minor issues;

1. language issues exist

2.  AI is also used for fog/fronts etc but not considered as important event, please see 2 papers:

a) Gultepe et al 2007; PAAG review paper on fog

b) MDPI applied sciences, Shan, Y. , I. GUltepe et al 2019, 9, 4487.

3. some sentences need to be rewriting. Please check your text.

Author Response

Overall, good topic is chosen to be studied. Authors tried to do their best but significant issues exist and need to be improved before going to printing/accepting.

Decision: major revisions

major points:

  1. front needs to be defined and why this paper doesn't follow a front definition by AMS/WMO

Several descriptions of front characteristics were already included in the initial version of the manuscript. The general idea was to emphasise the plurality of approaches, however, regarding Reviewer suggestion, the definition proposed by AMS (Glossary) has been introduced in the revised text. 

  1. why the parameters selected for front discrimination and AI analysis

For the initial stage of the analysis all parameters available  for chosen pressure levels were taken into account as well as some additional near surface parameters. They have been indicated as the most often used iu weather forecast preparation.

  1. why they are important, method is not clear.

The importance of those parameters results from Random Forest method, which indicated the parameters most important in the process of model training, as well as those to be neglected.

  1. results are given for extended frontal regions but for same no precip, cloud water content, and wind are provided, this is very weak point for this work

Thank you for paying attention to this point. We have improved the methodology description to be more clear. . Additional sentence has been added in Section 3.2:

For every point from the digitized database we add additional points in every direction, to artificially expand front size, and the same procedure was done for ERA5 data (both for surface and pressure levels fields)

  1. some figs are note indicated correctly, check.

Thank you for pointing this problem out. Figures now are in the correct order. 

  1. conclusions are very weak and not given in an itemized way.

We tried to correct our conclusions and made them in more itemized way.

Minor issues;

  1. language issues exist

The revised version of the manuscript has been corrected by a native speaker. 

  1. AI is also used for fog/fronts etc but not considered as important event, please see 2 papers:
  2. a) Gultepe et al 2007; PAAG review paper on fog
  3. b) MDPI applied sciences, Shan, Y. , I. GUltepe et al 2019, 9, 4487.

Thank you for your suggestion. We have browsed through those interesting papers and included the findings in our manuscript

  1. some sentences need to be rewriting. Please check your text.

The revised version of the manuscript has been corrected by a native speaker. 

Round 2

Reviewer 1 Report

After the detailed explanations and efficient clarifications provided by the authors in my comments and the respective comments of the two other reviewers, the quality of the paper has been significantly improved. During my first revision I had many doubts for the usefulness of this method, not only because of the deficiencies in the previous draft, but also because of my extensive employment with the climatology of fronts using the traditional methods. Nevertheless, the clarifications given by the authors as answers to the comments of the two other reviewers, helped me to read the manuscript with a modified philosophy.

Reviewer 3 Report

I am satisfied with corrections. Happy to see paper is improved.